# Are We Approaching Peak Meat Consumption? Analysis of Meat Consumption from 2000 to 2019 in 35 Countries and Its Relationship to Gross Domestic Product

**DOI:** 10.3390/ani11123466

**Published:** 2021-12-06

**Authors:** Clare Whitton, Diana Bogueva, Dora Marinova, Clive J. C. Phillips

**Affiliations:** 1School of Population Health, Curtin University, Perth, WA 6102, Australia; clare.whitton@postgrad.curtin.edu.au; 2Curtin University Sustainable Policy (CUSP) Institute, Curtin University, GPO Box U1987, Perth, WA 6845, Australia; diana.bogueva@curtin.edu.au (D.B.); d.marinova@curtin.edu.au (D.M.); 3Centre for Advanced Food Engineering (CAFE), Sydney University, Sydney, NSW 2006, Australia

**Keywords:** meat consumption, poultry, pork, beef, lamb, GDP, COVID-19, sustainability

## Abstract

**Simple Summary:**

Meat production has been associated with environmental degradation, yet at the same time many countries are increasing their consumption. We analyzed which countries are increasing and which decreasing their consumption of all the major meats consumed over a period from 2000 to 2019. There is evidence of peak meat consumption having been reached in several countries, but also evidence of continued consumption increases in many of the emerging-economy countries, probably due to greater affordability. We also found evidence of this when we attempted to link Gross Domestic Product (GDP) per capita to meat consumption per capita. In many emerging-economy countries, there was a direct link over the years studied, but in the higher income countries there was no relationship. We discuss how this observed dichotomous relationship will shape future trends in meat consumption.

**Abstract:**

Growing prosperity, but also disease outbreaks, natural disasters, and consumer preferences are changing global meat consumption. We investigated the 2000–2019 trends in 35 countries monitored by the Food and Agriculture Organization and the Organisation for Economic Co-operation and Development. We also tested relationships with Gross Domestic Product (GDP). Several countries appeared to be reaching peak consumption of some meats, and three (New Zealand, Canada, and Switzerland) have reached this. Poultry consumption increased over time in most countries, and beef and mutton/lamb consumption decreased in many. Using cluster analysis, we divided countries into two clusters: one in which increases in GDP per capita matched increases in meat consumption; and a second one of nine countries, for which there was no association between per capita change in GDP and meat consumption. There was evidence of a tipping point around USD 40,000 of GDP per capita, after which increases in economic well-being do not lead to increased meat consumption.

## 1. Introduction

Development has conventionally resulted in increased meat consumption. World meat consumption has quadrupled since 1961 in absolute and per capita terms [1], although COVID-19 may reduce global availability [2]. The growing human population has encouraged expansion of livestock, with 80 billion animals slaughtered annually to produce 340 million tonnes of meat for human consumption [1]. While meat consumption is positively associated with income [1], an early study of 120 countries identified that at a certain level of income per capita, total meat consumption decreases with income [3], but it did not analyze individual types of meat.

Climate change and environmental deterioration motivate people to reduce meat consumption [4]. Animal welfare and human health considerations, including cancers, obesity, zoonotic diseases, and potential loss of antibiotics, trigger rising flexitarianism and more plant-based meat alternatives [5]. Substitution of red meat for poultry [6] is also occurring, possibly because chicken is perceived as healthier, better for the environment and cheaper [7]. 

Whilst many factors shape consumer preferences, it is increasingly recognized that livestock has a major impact on ecosystems worldwide. Producing only 18% of the global food supply, livestock currently uses 77% of all agricultural land [8]. From small beginnings in subsistence agriculture, livestock production has witnessed continuous technological innovation and intensification, particularly in industrialized countries, with new genotypes and intensive feeding regimes to increase yields over ever-shorter time periods. Grains are diverted from direct human consumption to feeding animals [9], supporting an inefficient use of resources. In the process of two-stage nutrient digestion, first in animals and then humans [10], more than 90% of the proteins are lost. 

Furthermore, modeling by the Intergovernmental Panel on Climate Change [11,12] suggests that achieving the Paris Climate Accord is unrealistic with growing emissions from livestock. Ruminant meat has greenhouse gas effects 20 to 100 times higher than those of plant-based foods [13], with beef being the worst for the environment [14]. Livestock is also highly vulnerable to extreme weather events, including droughts, wild fires, and floods.

Whilst eating is a personal choice, subject to access, availability, and affordability, the implications for the common resources of land, air, water, biodiversity, and climate are proportionately greater than those of energy, transportation, buildings, and any other industry over a 20-year horizon. We hypothesized that this is inducing a shift towards less meat consumption. Hence, we searched for evidence for ‘peak meat’ in any countries–a point in time when consumption of animal meat, and beef in particular, peaks followed by a voluntary reduction in consumption [15], and whether shifts in meat consumption are linked to changes in Gross Domestic Product (GDP).

## 2. Materials and Methods

Below is a description of the data used in the study followed by an explanation of the statistical analyses conducted. As this research study did not involve human or animal participants, no ethical approval was necessary or sought.

### 2.1. Data and Indicators

We aimed to investigate a diverse selection of countries, covering the main markets globally. The Organisation for Economic Co-operation and Development (OECD) monitors agricultural statistics of both OECD countries, and key emerging economies. Annual estimates of consumption of beef and veal, poultry, pork, and sheep meat (in kg retail weight per capita, kg/cap) were downloaded for each year between 2000 and 2019 inclusive from the OECD-FAO Agricultural Outlook database [16]. Fish (including fish, crustaceans, mollusks, and other aquatic animals) were excluded in this analysis, because wild-caught fish, representing a substantial proportion of the fish consumed worldwide [17], do not have the conventional production processes of terrestrial animals. The descriptions of the different types of meat are presented in Table 1. Carcass weight to retail weight conversion factors used in constructing the data were: 0.70 for beef and veal, 0.78 for pigmeat, and 0.88 for both sheep meat and poultry meat [16]. Data were available for 35 countries (see Table 2). 

Data on total world consumption were also downloaded. Total meat consumption was calculated by summing the individual meat categories for each country. Percent contribution of each meat type to total meat consumption was calculated as: (consumption of each meat type, kg/capita, ÷ total meat consumption, kg/capita) × 100. 

The association between meat consumption and people’s economic situation was examined using annual estimates of GDP/capita (in current USD/capita), downloaded for each year between 2000 and 2019 inclusive, from the World Bank Group [19]. As the most commonly used indicator for comparing economic performance between countries, GDP per capita is the sum of marketed goods and services that are produced within a country, divided by the number of residents of that country. 

The data are publicly available and downloadable from the respective databases. All currency-related data are presented in US dollars (USD).

### 2.2. Statistical Analysis

For each meat type and each country, linear regression models of changes in consumption over time were constructed, regressing meat consumption against time (year), to detect significant increasing, decreasing, or static trends in consumption over time. Separate models were constructed for kilograms per capita of each meat type, and for percentage contribution of meat type to total meat consumption. Quadratic regression models were also tested, as well as straight-line relationships, inclusive of intercepts. To test our hypothesis that change in GDP per capita was associated with changes in consumption of different meat types, we constructed linear regression models of changes in GDP per capita over time (years) for each country. Then, the association between change in GDP per capita and change in meat consumption per capita was determined using linear regression models. As the people’s baseline economic status in the year 2000 was considered to potentially influence change in consumption, the associations between GDP per capita in the year 2000 and changes in meat consumption were also assessed by linear regression models. Both change in GDP per capita and GDP per capita in the year 2000 variables were transformed using log10 prior to analysis to improve normality of the residual distributions. 

Grouping of countries with different relationships between meat consumption and GDP was examined using cluster analysis to establish similarity in countries’ meat consumption and GDP changes. Ward’s method of hierarchical cluster analysis was used to produce a similarity measure of squared Euclidean distance [20]. The number of clusters was first determined by examining distances between clusters using a dendrogram and agglomeration schedule. Univariate linear regression models were then constructed for each country cluster, by regressing the change in meat consumption against, first, the change in GDP per capita, and, second, the GDP per capita in year 2000. All statistical analyses were conducted in IBM SPSS Statistics 26 (IBM Corp, Armonk, NY, USA). In terms of statistical significance, *p* values less than 0.05 were considered to be statistically significant.

## 3. Results

There were major changes in meat consumption during the 2000–2019 period, detailed below. In 2019, poultry was the most popular meat worldwide, followed by pork, then beef, and finally sheep and goat meat (Table 3). 

### 3.1. Meat Consumption

World meat consumption per capita increased between 2000 and 2019 (29.5 kg vs. 34.0 kg), by 0.34 (SE 0.03, *p* < 0.001) kg/capita/year (Appendix A). In most of the countries studied (26 of 35), total meat consumption per capita increased significantly over time (Figure 1, Table 4). The most substantial increases, by close to 2 kg/capita/year, were observed in countries with consumption levels below the 2000 world average, such as Russia, Vietnam, and Peru. In South American countries (Brazil, Argentina, Chile, and Colombia), which had relatively high consumption in 2000, annual increases by over 1 kg/capita were also observed. Overall decreases in total meat consumption were seen in six countries, most notably, New Zealand (86.7 kg/capita in 2000 to 75.2 kg/capita in 2019) and Paraguay (53.5 kg/capita in 2000 to 39.5 kg/capita in 2019) (Figure 2 and Appendix A).

#### 3.1.1. Beef

The contribution of beef to world total meat consumption was less in 2019 than in 2000 (18.9% vs. 22.8%) (Appendix A). Fifteen countries experienced absolute decreases in per capita beef consumption between 2000 and 2019, ranging from 12 kg/capita in New Zealand and Paraguay (around 0.5 kg/year), to less than 1 kg/capita in Thailand, Nigeria, and India (Appendix A).

Beef consumption as a proportion of total meat consumption increased only in Ethiopia, Israel, Saudi Arabia, Turkey, and Vietnam (Appendix A). In seven countries (including China, Indonesia, Pakistan, and UK), no change in the proportion of beef to total meat consumption was observed. Apart from Chile, Israel, and Kazakhstan, countries with an increasing trend in per capita beef consumption tended to have levels below the world average in 2000 and remained below this level in 2019. A decreasing linear trend in beef as a proportion of total meat consumption was observed in a majority of the studied countries (23 of 35) between 2000 and 2019 (Appendix A). The greatest reductions in percentage contribution between 2000 and 2019 were in Colombia (−30.5 percentage points), Ukraine (−30.2 percentage points), and India (−28.9 percentage points). 

#### 3.1.2. Pork

Per capita pork consumption increased slightly in China (24.0 kg in 2000 vs. 24.4 kg in 2019) and raised substantially in Vietnam (13.6 kg in 2000 vs. 26.0 kg in 2019) (Appendix A). Nineteen countries experienced increasing linear trends of varying magnitudes in pork consumption per capita, while seven countries experienced decreasing linear trends. In countries with decreasing trends in pork consumption per capita, the change was small in magnitude, except Canada which had a substantial decrease, from 22.6 kg/capita in 2000 to 16.3 kg/capita in 2019. The contribution of pork to world total meat consumption was lower in 2019 than in 2000 (32.6% vs. 38.6%) (Appendix A). A decreasing linear trend in pork as a proportion of total meat consumption was observed in 15 countries (Appendix A). In Vietnam and China, pork contributed two thirds of total meat consumption per capita in 2000, but by 2019 its contribution was only half of total meat consumption. 

#### 3.1.3. Poultry

Absolute poultry consumption per capita more than doubled in 13 countries (Appendix A). Nearly all countries studied (30 of 35) experienced a linear increasing trend between 2000 and 2019 in per capita poultry consumption (Appendix A). In 15 countries, the increase was greater than 10 kg/capita, and it was greater than 20 kg/capita in Peru, Russia, and Malaysia. World poultry consumption per capita was 14.8 kg in 2019, compared with 9.8 kg in 2000. A slight decrease in per capita poultry consumption was observed in Ethiopia, Nigeria, and Paraguay, while no trend was observed in Israel and Thailand. In contrast to the declining proportional contribution of beef and pork to world total meat consumption, the contribution of poultry increased from 33.0% to 43.4% between 2000 and 2019 (Appendix A). 

#### 3.1.4. Sheep Meat

A decreasing linear trend in sheep meat consumption was observed in a majority of countries (19 of 35) (Appendix A). As with beef consumption, countries experiencing increasing linear trends in sheep meat consumption were those with low consumption levels in 2000. An exception was Kazakhstan with a relatively higher sheep meat consumption which increased from 6 kg/capita to 8 kg/capita between 2000 and 2019. World sheep meat consumption in 2000 was 1.6 kg/capita, although New Zealand (25 kg/cap) and Australia (16 kg/capita) were outliers with high per capita consumption. These two countries experienced particularly large decreases of −1.31 (SE 0.13, *p* < 0.001) kg/capita/year and −0.44 (SE 0.04, *p* < 0.001) kg/capita/year, respectively. Sheep meat consumption was below 7 kg/capita in all other countries in 2000 and by 2019, it was < 9 kg/capita in all countries studied. A reduction was observed in the contribution of sheep meat to world total meat consumption (5.6% in 2000 vs. 5.1% in 2019) (Appendix A). 

### 3.2. Associations with GDP

Hierarchical cluster analysis dendrograms and agglomeration schedules for total meat (Appendix A and Appendix A) and for each meat type showed an inconsistently large increase in the measure of dissimilarity between clusters in the grouping of all countries. Two clusters were indicated (Figure 3) that were largely similar across meat type, except for UK, Korea, and Saudi Arabia which varied by meat type in the cluster they were assigned to (Appendix A). Cluster 1 comprised mostly countries with relatively small increases in GDP per capita over time and increases in meat consumption. It covered 26 countries. Cluster 2 comprised countries with greater changes in GDP per capita and variable (positive or negative) changes in meat consumption. For total meat consumption, Cluster 2 comprised Australia, Canada, Switzerland, UK, Israel, Norway, New Zealand, Saudi Arabia, and USA. Figure 3 displays a scatterplot of the relationship between per capita change in GDP and in total meat consumption. Among Cluster 1 countries, there was a significant association between per capita change in GDP and change in total meat consumption, with an increase of USD 9795 (SE 2148, *p* = 0.006) per year associated with a 1 kg/capita/year increase in total meat consumption (Table 5). The constant was USD 25,003. There was no association between change in GDP per capita and change in total meat consumption in Cluster 2 countries (*p* = 0.84). 

## 4. Discussion

We found poultry to be the main driver of increasing total meat consumption, while beef and sheep meat consumption generally decreased. Increase in GDP was associated with change in total meat consumption in many countries, but not in certain high-income countries. These trends probably reflect consumer appetite and historical technological industrialization. The 1950s were dominated by pork and beef, followed by poultry, which remained stable until 1980 [9]. The amount of beef produced globally increased dramatically between 1950 and 1990, with no significant expansion afterwards, while poultry surged, overtaking beef in 1997, mainly due to greater grain-conversion efficiency [9]. Our study shows that this substitution of poultry for beef has continued over the last two decades. Increased consumption of poultry is further driven by population growth and rising incomes in developing countries [7,21]. 

Despite growing meat consumption in low- and middle-income economies, disparities between countries remain. Some countries consume less than 5 kg/capita/year (notably India and Ethiopia). Some are as high as 100 kg/capita/year (notably Australia and USA), which equates to 25 kg protein [22], higher than the recommended dietary intake for protein. The EAT-Lancet Commission on Healthy Diets from Sustainable Food Systems [23] reports that a protein intake of 0.8 g/kg bodyweight is adequate, or 20.4 kg for a 70 kg person. It also suggests that red meat be avoided altogether in the diet. 

### 4.1. Changes in Meat Consumption 

Projections by the UN for global meat production created within our chosen time period forecasted an increase by 2050 [24]. However, there are now trends for some countries in the opposite direction, namely, New Zealand, Canada, Switzerland, Paraguay, Nigeria, and Ethiopia (Table 1, Appendix A and Appendix A). The reasons for this vary. In the case of New Zealand, Canada, and Switzerland, vegetarianism has been on the rise [25,26,27]. A larger proportion of the population is now consciously decreasing its meat consumption because of environmental and health concerns, challenging existing cultural associations [28]. Improved availability of alternative proteins is also behind the reduction in meat consumption [29] and so too are animal welfare, ethics, concerns about the environment, sustainability of food systems, and health [27]. The situation in the remaining three countries is different, with reduction in meat consumption because of economic unaffordability. Paraguay is one of the world’s top 10 beef-exporting countries with the highest rates of deforestation linked with cattle grazing [30], where domestic meat consumption competes with export opportunities and severe droughts and wildfires impact production. In Nigeria, most demand for meat is met by pastoralists who use traditional techniques and are impacted by weather calamities. Decrease in per capita meat consumption in Nigeria and Ethiopia is also driven by population growth with meat consumed only on special occasions [31].

In some countries, substantial fluctuations appeared in consumption between 2000 and 2019. For example, pork witnessed major fluctuations as the preferred meat in China. In 2007, pork consumption dropped, which can be attributed to the >50% increase in price following outbreaks of swine influenza, and the SARS outbreaks in humans at that time [32]. We found that in China, while pork consumption increased, the contribution of pork to total meat consumption decreased, replaced by sheep and poultry. In 2015–2016, the Chinese market experienced a reduction in pork production as domestic consumers became more interested in other meat sources, including beef and chicken [33], with concerns about safety [34] and healthy diets [35]. Fluctuations were also noted in New Zealand and Paraguay (Figure 2). In the case of New Zealand, despite the overall reduction in meat consumption over the time period studied, there was a relatively small increasing trajectory in total meat consumption beginning in 2010, attributable to increases in poultry consumption. This increase followed a period of low meat consumption in the previous decade, which may have been an aftermath of the bovine spongiform encephalopathy (BSE or mad cow) crisis. In the case of Paraguay, the peak in 2011 may have been due to an outbreak of foot-and-mouth disease, which resulted in export bans. 

A reduction in sheep meat consumption was observed in most countries. An exception was Kazakhstan, which experienced the greatest absolute increase in sheep meat consumption, a highly preferred meat option associated with the growing affluence in that country. Kazakhstan has been intensifying its rangeland production of sheep meat, with fertilizer programs and other measures to increase production and self-sufficiency [36]. 

### 4.2. Changes in Meat Consumption with GDP

Changes in meat consumption were either unrelated to GDP – for a group of nine countries (eight OECD countries plus Saudi Arabia, which is a G20 country), or positively related to change in GDP. In the latter group, for every USD 9795 increase in GDP/capita/year, there was a 1 kg increase in total meat consumption/capita/year. Clearly, only a small proportion of additional GDP was spent on meat, with even the most expensive meat, beef, averaging only about USD 5/kg [37].

The Cluster 2 nations, for which we observed that changes in meat consumption were unrelated to GDP, may have passed a ‘tipping point’ – that is, a point at which GDP increases no longer lead to growth in meat consumption. These countries all had higher 2019 GDPs (Table 6) than countries in which meat consumption was positively related to change in GDP. All these countries are considered high-income in absolute and relative terms and have traditionally held meat in high respect. The observed tipping point signifies a substantial change in people’s attitudes towards meat.

The highest GDP among Cluster 1 countries, in which meat consumption was positively related to change in GDP, was Japan at USD 40,247. This suggests that at a GDP per capita exceeding about USD 40,000, growth of GDP is no longer a driver of growth in meat consumption. This aligns broadly with an earlier study which reported USD 35,035 (as of 2005) as the GDP turning point for a reduction in meat consumption [3]. Our study adds to the literature that these patterns are consistent across types of meat, as indicated by the results of the cluster analysis.

Of the countries that were reducing their meat consumption in the 2000–2019 period, we hypothesize two types of reasons. First, the countries Ethiopia, India, Nigeria, and Paraguay have seen little or no growth in GDP and have growing populations, hence, there is less meat available per capita. People in Paraguay experienced the biggest reduction in meat consumption, particularly beef and, to a lesser extent, poultry, due to export orientation and weather calamities. Second, the countries Canada, New Zealand, and Switzerland may have reduced consumption because of the contribution that meat production makes to global warming, reduced food security, animal welfare, and animal ethics [27]. It is official government policy to restrict the consumption of animal products in Canada (red meat), New Zealand, and Switzerland [39]. However, Canada also has a healthy-eating strategy, and although it makes it clear that changes are needed in saturated fat intake, there is no explicit mention of meat [40]. This may be due to government lobbying. Of the ten most influential lobby groups in Canada, two are representatives for the cattle and poultry industry, and a third is an animal welfare advocacy group [41]. Hence, it can be assumed that government officials in Canada are under pressure from both sides regarding the future of the meat industry.

### 4.3. The Future of Meat Consumption

Despite substantial scientific evidence about the disadvantages of excessive meat consumption for human health, environmental impacts, and animal welfare [42], an overall decrease in people’s meat intake is not yet evident, other than beef and sheep–the choices with the highest environmental footprints. Future meat consumption will be shaped by new trends in supply and demand.

The FAO estimates that meat production would have declined by 1.7% in 2020, due to animal diseases affecting poultry and pork industries, the effects of droughts, and COVID-19 [43]. However, a forecast by OECD-FAO [44] anticipated a strong growth to 2028 with no explanation of why the two constraining factors of diseases and climate change should not continue to limit meat production. Since the OECD-FAO projection, the impact of COVID-19 on the global economy has emerged as an additional dominant factor affecting consumption and trade. If the reduction is continued in 2021, it will be an unprecedented sign of peak meat consumption having been reached, representing the start of a more durable change.

Beef production also changed in 2020 due to COVID-19 [45]. A downward trend is evident in 2021 in China and USA [46]. Furthermore, COVID-19 may impact demand if consumers opt to avoid beef because of zoonotic potential. Weather extremes may also influence beef production, such as the 15% decline in Australia during the 2019 drought [47].

On the demand side, we may witness further reduction in meat consumption because of people’s choice, as already seen in the Cluster 2 countries. Despite clear concerns about climate breakdown viewed as an issue of a global emergency by two thirds of 1 million people surveyed from 50 countries, switching to plant-based dietary choices was not seen positively as the solution [48]. In a recent simulation study [49], Henchion et al. reported the projected demand for animal-based protein until 2050 and noted that decreases in meat consumption in European and American regions were possible, should effective policies be implemented to support a more sustainable food system. However, the study reported that these decreases would be accompanied by increased consumption in African and South-East Asian regions, rendering little change to the global average [49].

There is, however, a game changer emerging in the meat supply-and-demand plot. It relates to non-livestock-based alternatives, such as cultured meat and plant-based alternatives. While evidence is yet to emerge of these types of meat disrupting the conventional animal-based proteins, analysts predict that cultured meat and the novel vegan meat replacements have the potential to create 60% of the market by 2040 [50]. The global plant-based protein market is already growing rapidly [51,52] and China is likely to become the new frontier for meat substitutes [53]. 

### 4.4. Limitations of the Study

The observed positive relationship between GDP change and meat consumption change could have been different had data from a different set of countries been used. These 35 countries are not necessarily a representative sample of all countries. The GDP per capita metric does not account for within-country inequality, and skewed distributions of GDP within countries exist. This may have resulted in the attenuation or inflation of the observed associations between changes in GDP and meat consumption. Meat consumption estimates are based on meat available for consumption, but does not account for waste at either the individual or retail level. The amount of waste likely varies between countries, which may have caused non-random error across estimates.

Fish were excluded from our study, and it is important to remember that consumer preferences globally have shifted not only towards higher consumption of poultry, but also fish. The oceanic fish catch and fish-farm production have continued to grow in recent years and reached an all-time record of 178.5 million tonnes in 2018 [54]. Driven by human appetite for seafood, four fifths of marine fisheries are being fished at or beyond their sustainable capacity, resulting in a decline of many species [9,54]. The world now produces more seafood from fish farms than wild catch and, like meat from terrestrial animals, the global production of fish and seafood quadrupled in the last 50 years [55]. Further analysis of fish consumption can shed light on how people’s food preferences are changing.

We focused our attention on explaining changes in meat consumption and GDP, but there are many other factors that influence meat consumption, such as culture, religion, gender and identity, market structures, and geographic factors. These were beyond the scope of our study but could usefully be included in future investigations. 

## 5. Conclusions

Meat consumption varies considerably between countries, the majority of which have increased consumption between 2000 and 2019. As well as increasing total consumption, there has been substitution of poultry for the other meats. There are specific instances of individual meats declining, for example pork in China, as a result of disease concerns. A few countries are reducing total meat consumption, and there is evidence that some types of meat are approaching peak meat consumption in some countries. Other countries are likely to follow. We also found evidence of a tipping point around USD 40,000 of GDP per capita, after which increases in economic well-being within a country do not lead to growth in meat consumption.

We analyzed a twenty-year retrospective period of relative economic stability which also generated major environmental impacts. The period ahead is likely to be very different. While the world struggles to deal with COVID-19, the environmental and climate emergency is likely to shift the food production model towards better ways of providing protein for the growing global population. The food systems will need to be re-shaped to support healthy diets and align with sustainable development. These new ways of producing food are likely to be less exploitative of nature and the animal world, as the human population transitions to better relationships with the biophysical world on which it depends.

## Figures and Tables

**Figure 1 animals-11-03466-f001:**
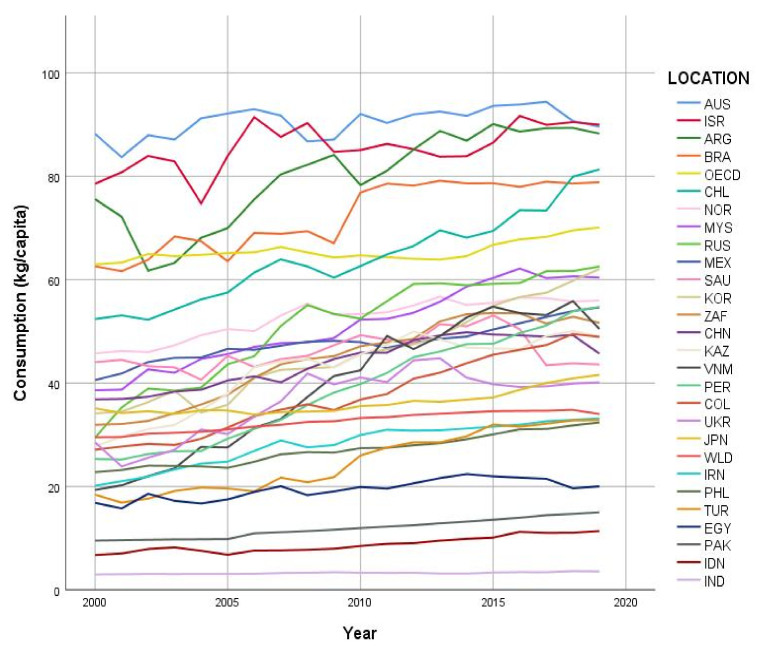
Total meat consumption over time (years) in countries with increasing consumption, for countries with statistics documented by the Organisation for Economic Co-operation and Development–Food and Agriculture Organization of the United Nations (OECD–FAO) database [16].

**Figure 2 animals-11-03466-f002:**
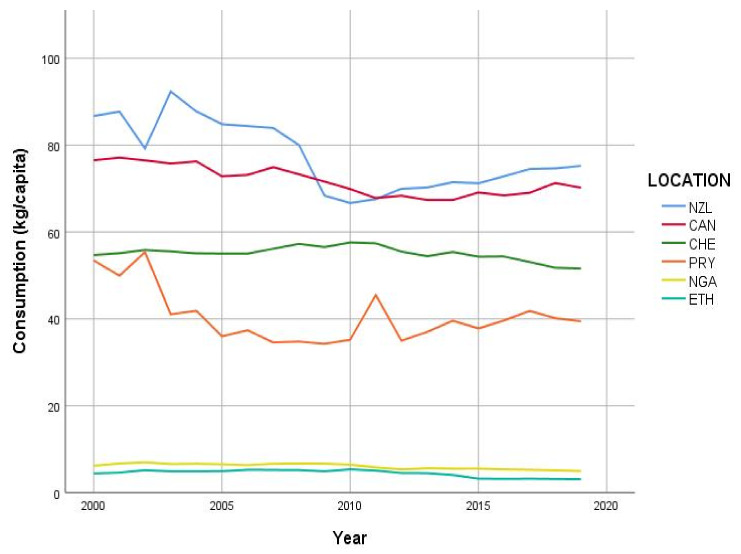
Total meat consumption over time (years) in countries with decreasing consumption, from statistics documented by the Organisation for Economic Co-operation and Development (OECD)–Food and Agriculture Organization of the United Nations (OECD–FAO) database [16].

**Figure 3 animals-11-03466-f003:**
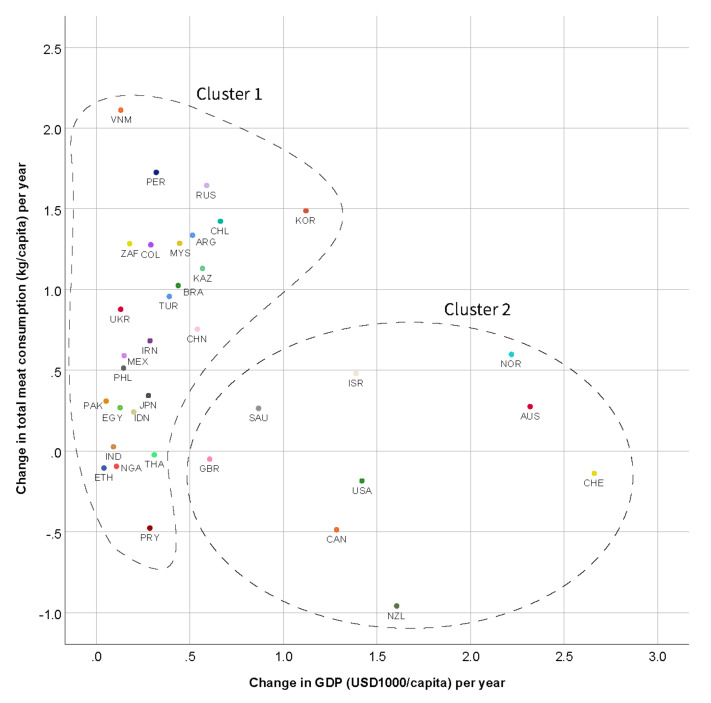
Scatterplot of change in Gross Domestic Product (GDP) (USD 1000/capita) per year by change in total meat consumption (kg/capita) per year. Circles indicate country clusters.

**Table 1 animals-11-03466-t001:** Description of meats analyzed in the study.

Meat	Description
Poultry	Thighs, wings, breast, ribs, and back of domestic fowls (chicken), guinea fowl, ducks, geese, and turkeys
Beef and veal	Carcass weight of meat from cattle, excluding the hide or skin, the head where it joins the spine, the fore feet at the knee joint, the hind feet at the hock joint, the large blood vessels of the abdomen and thorax, the genito-urinary organs (other than the kidneys), offal, and the tail
Pig meat	Includes pork and bacon but excludes the genito-urinary organs (other than the kidneys) and offal
Sheep meat	Sheep meat excluding skin, offal, genito-urinary organs, and feet

Source: Food and Agriculture Organization of the United Nations (FAO) [18].

**Table 2 animals-11-03466-t002:** Countries included in the study.

Country	Abbreviation	Country	Abbreviation
Argentina	ARG	Mexico	MEX
Australia	AUS	Malaysia	MYS
Brazil	BRA	Nigeria	NGA
Canada	CAN	Norway	NOR
Switzerland	CHE	New Zealand	NZL
Chile	CHL	Pakistan	PAK
PR China	CHN	Peru	PER
Colombia	COL	Philippines	PHL
Egypt	EGY	Paraguay	PRY
Ethiopia	ETH	Russian Federation	RUS
United Kingdom	GBR	Saudi Arabia	SAU
Indonesia	IDN	Thailand	THA
India	IND	Turkey	TUR
Iran	IRN	Ukraine	UKR
Israel	ISR	United States of America	USA
Japan	JPN	Vietnam	VNM
Kazakhstan	KAZ	South Africa	ZAF
Korea, Republic of	KOR		

**Table 3 animals-11-03466-t003:** World meat consumption, 2019.

Type of Meat	Total Consumption (kg/capita)	Share
Poultry	14.7	43%
Pork	11.1	33%
Beef	6.4	19%
Sheep and goat	1.8	5%
Total	34.0	100%

Source of data: Organisation for Economic Co-operation and Development (OECD)–Food and Agriculture Organization of the United Nations database [16].

**Table 4 animals-11-03466-t004:** Trends in per capita meat consumption, 2000–2019.

Country.	Total Meat	Beef/Veal	Pork	Poultry	Mutton/Lamb
Argentina	↑	↓	↑	↑	↓
Australia	↑	↓	↑	↑	↓
Brazil	↑	NS	↑	↑	NS
Canada	↓	↓	↓	↑	NS
Chile	↑	↑	↑	↑	↓
China (People’s Republic of)	↑	↑	↑	↑	↑
Colombia	↑	↓	↑	↑	↓
Egypt	↑	NS	↓	↑	↓
Ethiopia	↓	↓	↓	↓	↓
India	↑	↓	↓	↑	↓
Indonesia	↑	↑	NS	↑	NS
Iran	↑	↑	-	↑	↓
Israel	↑	↑	↓	NS	↑
Japan	↑	NS	↑	↑	↓
Kazakhstan	↑	↑	NS	↑	↑
Korea	↑	↑	↑	↑	↑
Malaysia	↑	↑	↓	↑	↑
Mexico	↑	↓	↑	↑	↓
New Zealand	↓	↓	↑	↑	↓
Nigeria	↓	↓	↑	↓	↓
Norway	↑	↓	↑	↑	↓
Pakistan	↑	↑	NS	↑	↓
Paraguay	↓	↓	NS	↓	NS
Peru	↑	↑	↑	↑	↓
Philippines	↑	NS	↑	↑	↑
Russia	↑	NS	↑	↑	↑
Saudi Arabia	↑	↑	↑	↑	↓
South Africa	↑	↑	↑	↑	NS
Switzerland	↓	↓	↓	↑	↓
Thailand	NS	↓	NS	NS	↑
Turkey	↑	↑	↓	↑	NS
Ukraine	↑	↓	↑	↑	NS
United Kingdom	NS	NS	↓	↑	↓
United States	NS	↓	NS	↑	↓
Vietnam	↑	↑	↑	↑	↑
World	↑	↓	↑	↑	↑

Note: ↑ = increase; ↓ = decrease; NS = no significant trend detected; - = pork was not consumed in Iran.

**Table 5 animals-11-03466-t005:** Univariate linear regression analysis of the association between year 2000 GDP (USD 1000/capita) and Gross Domestic Product (GDP) change with change in total meat consumption (kg/capita/year).

Country Cluster	Terms	Unstandardized Coefficients		*p* Value	R Square
		β	SE		
1	Constant	−0.398	0.810	0.63	
	GDP (USD 1000/capita) in year 2000	0.364	0.245	0.15	0.085
	Constant	1.398	0.232	<0.001	
	Change in GDP (USD 1000/capita) per year (Unstandardized β)	0.991	0.332	0.006	0.271
2	Constant	−1.71	3.808	0.67	
	GDP (USD 1000/capita) in year 2000	0.386	0.871	0.67	0.027
	Constant	−0.053	0.228	0.82	
	Change in GDP (USD 1000/capita) per year (Unstandardized β)	0.192	0.895	0.84	0.007
Dependent Variable: Change in consumption (kg/capita) per year (Unstandardized B)					
Country cluster 1 = ARG, BRA, CHL, CHN, COL, EGY, ETH, IDN, IND, IRN, JPN, KAZ, KOR, MEX, MYS, NGA, PAK, PER, PHL, PRY, RUS, THA, TUR, UKR, VNM, ZAF					
Country cluster 2 = AUS, CAN, CHE, GBR, ISR, NOR, NZL, SAU, USA					

Note: Change in GDP is represented as Log10 (change in GDP).

**Table 6 animals-11-03466-t006:** Cluster 2 countries, in which changes in meat consumption were unrelated to GDP.

Country	Abbreviation	Per Capita GDP in 2019	Per Capita GDP-PPP in 2019 *
Switzerland	CHE	USD 81,994	USD 68,628
Norway	NOR	USD 75,420	USD 63,633
United States of America	USA	USD 65,118	USD 62,530
Australia	AUS	USD 54,907	USD 49,854
Canada	CAN	USD 46,195	USD 49,031
Israel	ISR	USD 43,641	USD 40,145
United Kingdom	GBR	USD 42,300	USD 46,200
New Zealand	NZL	USD 42,084	USD 42,888
Saudi Arabia	SAU	USD 23,140	USD 46,962

Notes: GDP—Gross Domestic Product; PPP—purchasing power parity. * Source: The World Fact Book [38].

## Data Availability

Data are available from the Corresponding author on request.

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
