# Peer review of "Are We Approaching Peak Meat Consumption? Analysis of Meat Consumption from 2000 to 2019 in 35 Countries and Its Relationship to Gross Domestic Product"

_animals, 2021, doi:10.3390/ani11123466_

Round 1

Reviewer 1 Report

Authors analyzed which countries are increasing and which are decreasing their meat consumption over a period from 2000 to 2019 and they assert founding evidence of peak meat consumption having been reached in a number of countries, as well evidence of consumption increasing in many emerging economy countries.

One of the hypothesis tested in this paper is whether shifts in meat consumption are linked to changes in Gross Domestic Product. One of the sources used for this paper - Ritchie, H., & Roser, M. 2019 – also correlates higher meat consumption with Gross Domestic Product, or even national minimum wages.

Nonetheless, this manuscript could present a more clear justification on countries selection, and explain why some countries are included in cluster 1 and why others are in cluster 2. Assuming Brazil, Chile, Colombia are three countries (from Latin America) with high meat consumption rates, and therefore its selection is justifiable, why Switzerland is only selected one country from Europe? Why not Spain that presents the highest meat consumption rate in Europe, per capita?  (Ritchie, H., & Roser, M. 2019) Or even Portugal, with second highest consumption rate in Europe, per capita, (Ritchie, H., & Roser, M. 2019), with one of the lowest Gross Domestic Product in Europen Union?

The countries comprising in two clusters according to their Gross Domestic Product is presented as a accurate criteria. For instance, Cluster 2 comprised Australia, Canada, Switzerland, UK, Israel, Norway, New Zealand, Saudi Arabia and USA. If the criteria is Gross Domestic Product, why Japan, China or Russia don´t belong in Cluster 2?   

It is stressed in the Discussion that these meat consuming trends probably reflect consumer appetite and historical technological industrialization. I encourage the development of this idea. It would be relevant to add as an important factor the differences of livestock sector comparing capitalist countries with less developed countries. Especially in the most developed countries, there has been an intensification of livestock industry propelled by capitalism - high income for investors with low processing animal costs, cheap labor, less animal welfare, extinction of small farmers and more monopoly to bigger companies (power concentration, etc). Plus, due to lobbies, US, Canada, Brazil, European countries, and others, meat producers do have financial support, a major factor that contributes to lower meat prices and consumption increasing. When confronting meat consumption disparities, or triggers, – market, religion, culture and narratives, gender and identity, and geographic characteristics (i.e. intensive or extensive farming) could also at least be mentioned as other important factors in the Discussion section.

Author Response

File attached contain responses to reviewer 1

Reviewer 2 Report

The authors analysed trends in meat consumption in 35 countries over the period 2000-2019 using food availability data from the FAO. Consistent with other research, the analysis found a significant increase in poultry but a decline in most countries in beef and lamb consumption. They used OECD data on GDP per capita to find correlations between rising national wealth and increases in meat consumption, although this was not found among higher income countries.

The research is timely as the role of meat production and consumption is finding the spotlight more and more amid pressure to address the global climate crisis. While the links between animal sourced foods and environmental degradation are becoming clearer, there are still many questions about trends in consumption and how to promote a diet that will help keep us within planetary limits. This paper provides a useful longitudinal and geographically diverse analysis of trends and links to economic indicators.

The paper was written well and the authors set out their objectives and methods clearly. As a general observation, I think the paper could benefit from including references to commonly cited scientific reports calling for reductions in meat consumption (e.g. the 2019 EAT Lancet Commission). This will also help provide some context to the kg/capita figures mentioned throughout.

The countries used for the analysis included most of the highest meat consuming nations and came from multiple continents. While the list of countries was a reasonable set, the authors rightly note in the limitations that “These 35 countries are not necessarily a representative sample of all countries”. But despite this, the authors frequently refer to their results as reflecting “worldwide” consumption (.e.g Line 136 and Table 3 World meat consumption). I would suggest looking at some rewording to frame the findings as reflecting the 35 countries included in the research.

Below are further specific comments on the paper:

Line 81

Fish (including fish, crustaceans, molluscs and other aquatic animals) were excluded in this analysis, because wild-caught fish representing 54% the fish consumed worldwide [17] do not have the conventional production processes or environmental impact of terrestrial animals.

Line 380

The world now produces more seafood from fish farms than wild catch…

  • The fish facts above seem to contradict, one saying wild catch dominates while the later comment says aquaculture production is now the most common.
  • I wouldn’t agree with the statement that fish production is not linked to the same environmental impact as terrestrial animals. Greenhouse gas emissions linked to aquaculture can be in the same range as those for poultry and pork, whereas eutrophication and scarcity weighted freshwater withdrawals are often higher than poultry. See, for example:
    Poore, J., & Nemecek, T. (2018). Reducing food’s environmental impacts through producers and consumers. Science, 360(6392), 987-992.

Figures 1&2

  • It would be good to add the full country names as it was confusing to identify some of the countries. For example, many readers won’t be familiar with CHE referring to Switzerland. If the full names aren’t feasible, then at least longer abbreviations would be helpful.

I think if Figure 2 is to be included, there should be some text added that comments on the potential reasons for the spikes seen with New Zealand and Paraguay, in particular. Were there differences in how data were collected at these points? Were there food safety scares or pricing issues that might have contributed to these highs/lows? Also, NZ appears to be on a pretty steady increasing trajectory over the last decade which should be noted as well. I know there are some country-specific observations later in the paper, so it may just be a matter of moving some of these points up in the text.

Table 4. Trends in per capita meat consumption, 2000-2019

  • I had to read through this table a few times before it was clear what it was telling me. I wonder whether there’s a clearer way to present the information. Perhaps you could include a table with all of the countries on the left and the columns could have headers such as “total meat consumption”, “beef”, “pork”, “poultry”, etc. and show a symbol for each country indicating the trend, e.g. ­ ¯  °. This would allow the reader to quickly scan for each country and see the variations in trends by meat and total consumption.

Line 237

Note: Change in GDP is represented as Log 10 (change in GDP

  • Missing parentheses.

Line 254

higher than the recommended dietary intake for protein.

  • As mentioned above, this would be a good place to include a reference to international reports such as the 2019 EAT Lancet Commission and what their annual per capita consumption recommendation is.
    Willett, W., Rockström, J., Loken, B., Springmann, M., Lang, T., Vermeulen, S., ... & Murray, C. J. (2019). Food in the Anthropocene: the EAT–Lancet Commission on healthy diets from sustainable food systems. The Lancet, 393(10170), 447-492.
    Springmann, M., Clark, M., Mason-D’Croz, D., Wiebe, K., Bodirsky, B. L., Lassaletta, L., ... & Willett, W. (2018). Options for keeping the food system within environmental limits. Nature, 562(7728), 519-525.

Line 277

2003 pig high fever…

  • Does this refer to swine fever?

Table 6

Per capita GPD in 2019

  • Misspelling of GDP

Line 320

It is official government policy to restrict the consumption of animal products in Canada (red meat), New Zealand and Switzerland [37]. Canada has a healthy eating strategy, and although it makes it clear that changes are needed in saturated fat intake, there is no explicit mention of meat [38].

  • The references to Canada in these two sentences seem to contradict. I’m not aware of any policy in Canada on restricting animal sourced food consumption. The 2019 Canada Food Guide did make a shift to promote plant-based protein, but I don’t believe there’s ever been explicit messages about reducing meat consumption for the reasons that are mentioned later in your paragraph. I also didn’t find evidence for the first sentence in your citation [37].

Line 390

there is evidence that some meats approach a peak meat consumption,…

  • “some countries are approaching peak meat…..”?

Author Response

File attached contain responses to reviewer 2